

# iDILI-MT: identifying drug-induced liver injury compounds with a multi-head Transformer

Wanrong Zheng[1] and Fobao Lai[2]

[1] College of Medical Nursing, Minxi Vocational and Technical College, Longyan, Fujian, China
[2] Department of Oncology, Longyan First Affiliated Hospital of Fujian Medical University, Longyan, Fujian, China

## ABSTRACT

Drug-induced liver injury (DILI) is a leading cause of late-stage drug attrition and post-approval withdrawals, making early *in silico* risk assessment essential for drug safety. We present iDILI-MT (identifying drug-induced liver injury compounds with a multi-head Transformer), a self-contained computational framework that integrates a feed-forward network for sequential feature extraction, a multi-head Transformer encoder for contextual representation learning, and a squeeze-and-excitation attention module for channel-wise feature recalibration. Evaluated on a curated set of 1,919 small-molecule compounds, iDILI-MT outperformed traditional machine-learning classifiers and state-of-the-art graph neural networks, achieving a mean area under the receiver-operating-characteristic curve (AUC-ROC) of 0.8499, area under the precision-recall curve (AUC-PR) of 0.8905, and F1 score of 0.8173 across ten trials. Attention-weight analysis reveals that the combined multi-head and squeeze-and-excitation attention mechanisms effectively highlight key substructural and chemical motifs associated with hepatotoxicity. These findings indicate that iDILI-MT provides an useful method for detecting compounds at risk of DILI, potentially accelerating safety assessments in drug development.

# INTRODUCTION

## Background

Drug-induced liver injury (DILI) encompasses a spectrum of hepatic damage that can occur when people take medications, dietary supplements, herbal products, or other chemical substances (*Bell & Chalasani, 2009*; *Suk & Kim, 2012*). The liver's central role in drug metabolism makes it particularly vulnerable, as it processes most substances that enter the bloodstream, potentially leading to toxic accumulation (*Bissell et al., 2001*; *Lin et al., 2003*). Healthcare systems are facing significant diagnostic challenges with DILI as its symptoms are diverse, including elevated liver enzymes, jaundice, fatigue, and abdominal pain—often mimic other liver conditions. The variable onset time, ranging from days to months after starting medication, further complicates diagnosis and causality assessment (*Atienzar et al., 2016*; *Barnhill, Real & Lewis, 2018*; *Weber & Gerbes, 2022*). Besides, the unpredictable nature of individual drug metabolism poses major challenges in

Corresponding author
Fobao Lai, lfbjxgz@126.com

pharmaceutical development (*Thompson, 2001*; *Nicholson & Wilson, 2003*; *Park et al., 2011*; *He et al., 2024*). Today, modern prevention strategies employ advanced screening methods supported by artificial intelligence and machine learning to predict potential liver toxicity of drug compounds (*Nguyen-Vo et al., 2020*; *Lee & Yoo, 2024*). However, some forms of DILI only become apparent after widespread use, leading to drug withdrawals even after market approval (*Regev, 2014*; *Andrade et al., 2019*). Treatment focuses on immediate discontinuation of the suspected drug and supportive care (*Andrade et al., 2019*). While mild cases often resolve within months, severe cases may require intensive intervention or liver transplantation (*Neuberger, 2016*). Computational biologists continue to focus on developing better screening method for early detection and identifying drug candidates that may cause risks of DILI to improve both safety and efficiency in drug discovery (*Chen et al., 2014*; *Blomme & Will, 2015*; *Weaver et al., 2019*).

## Related work

Traditional approaches for screening DILI compounds had been mainly relied heavily on experimental animal testing. However, these methods face significant limitations, including high budget, time-consuming, and skilled experimenters. *Fraser, Bruckner & Dordick (2018)* highlighted that animal models not only require substantial time and resources but also frequently fail to accurately translate to human outcomes due to species-specific differences in drug metabolism. This translation gap was fairly discussed in a comprehensive analysis by *Shanks, Greek & Greek (2009)*. Their findings revealed that over 40% of drug toxicities observed in humans were found during animal studies, raising serious concerns about reliance on these models alone. While *in vitro* methods may offer alternative approaches, they struggle to replicate the complex interactions between different cell types and metabolic pathways that characterize human liver function (*Funk & Roth, 2016*; *Atienzar & Nicolas, 2018*; *Mirahmad et al., 2022*). To partially address these limitations, researchers have gradually designed more computational method for early identification of DILI compounds. *Ekins, Williams & Xu (2010)* employed Bayesian modeling and chemical fingerprints for early-stage DILI drug screening. However, their study was limited by modest performance metrics and a relatively small training dataset that inadequately captured DILI patterns. Their approach failed to fully address the complex biological mechanisms underlying liver injury, relying instead on a ligand-based methodology that oversimplified these pathways. Additionally, their work was constrained by dataset quality issues, limited diversity, and inherent class imbalance problems. Subsequent studies by *Zhang et al. (2016)* and *Ai et al. (2018)* expanded the algorithmic toolkit to include more sophisticated methods like random forests (RF), support vector machines (SVM), and gradient boosting (GB) techniques. *Zhang et al. (2016)* developed computational models to predict DILI compounds using substructural patterns on a dataset of 1,317 compounds. They found that the SVM model based on FP4 fingerprints had produced the most accurate model, achieving 75% accuracy on external validation. Their findings also identified six key molecular substructures associated with liver toxicity, including tertiary mixed amines, alkylthiols, and arylfluorides, which can serve as structural alerts during drug development. However, their study was limited by dataset

issues like class imbalance and annotation inconsistencies, while their chemical fragment-based approach oversimplifies DILI's complex biology, neglecting critical factors such as metabolic activation and genetic susceptibility. The model provides limited biological insights and lacks integration with pharmacokinetic data or biological pathways, making it less comprehensive than modern multi-data ensemble approaches. *Ai et al. (2018)* introduced an advanced computational approach for predicting DILI compounds by combining multiple machine learning algorithms with diverse molecular fingerprints. Their ensemble model, which integrated the five best-performing base classifiers, achieved an accuracy of 71.1% during cross-validation and impressively improved to 84.3% when tested on external compounds. Their model was proved to outperform previous computational methods in both accuracy and sensitivity, making it valuable early compound screening platform with a web server provided. *Ai et al.*'s *(2018)* model suffers from limited dataset representation, simple molecular fingerprints, poor specificity, and lack of interpretability due to its "black box" ensemble approach. These limitations compromise the model's reliability for drug development applications and highlight the need for more comprehensive data integration and transparent prediction methods. DeepDILI, proposed by *Li et al. (2020)*, identifies drug-induced liver injury by combining conventional machine learning algorithms with deep neural networks to create a model-level representation approach. The researchers implemented a temporal validation strategy, training their model on drugs approved before 1997 to test its ability to predict liver toxicity in newer compounds—a more realistic scenario for drug development workflows. Their analysis identified important molecular features associated with hepatotoxicity that could serve as structural alerts for further drug development. While achieving promising accuracy of up to 80.7%, the model faces several critical limitations: insufficient dataset diversity potentially causing bias, reduced generalizability to novel drug chemotypes not represented in the training data, and challenges with interpretability despite the hybrid approach attempting to balance predictive power with mechanistic insights. Most recently, *Lee & Yoo (2024)* introduced InterDILI, an interpretable machine learning approach for drug-induced liver injury prediction. The model enhances interpretability by integrating permutation feature importance analysis with attention mechanisms, addressing a critical need in computational toxicology. Evaluated across five public datasets, InterDILI demonstrated robust performance with area under the receiver operating characteristic curve (AUROC) values of 0.88–0.97 and area under the precision-recall curve (AUPRC) values of 0.81–0.95. These strong results enabled valuable contributions to early drug development by facilitating proactive identification of potentially hepatotoxic compounds and highlighting specific molecular substructures that could be modified to reduce DILI risk. Despite these achievements, the InterDILI model has several limitations. Its reliance on the DILIrank dataset introduces potential representation biases due to insufficient diversity and completeness in capturing the full spectrum of liver-toxic compounds. Although the model enhances interpretability, its mechanistic insights remain insufficient to fully elucidate the complex biological pathways underlying DILI. The emphasis on overall accuracy metrics overshadows clinically crucial

considerations of false positives and negatives, which directly impact drug development decisions.

## Motivations and objectives

Computational methods for DILI prediction face several significant challenges despite their promising results. The complex underlying mechanisms of DILI involve intricate molecular-biological pathway interactions that are difficult to model computationally. Limited training data availability constrains deep learning applications, while conventional machine learning approaches often fail to capture the subtle patterns associated with hepatotoxicity. Problems in molecular representation, such as bit collisions in fingerprint encoding, can obscure structural correlations critical to toxicity determination. These limitations highlight the need for more sophisticated methodological approaches to address this essential drug discovery challenge. Our research introduces iDILI-MT (identifying drug-induced liver injury compounds with a multi-head Transformer), a computational framework for DILI compound identification that employs multi-head Transformer architecture. This approach enhances conventional deep learning models through parallel self-attention mechanisms that process different levels of molecular structure. The transformer's multiple attention heads capture diverse biochemical interaction patterns, addressing both local substructural features and global molecular topologies simultaneously. We integrate molecular representations with positional encodings to preserve structural information, enabling better differentiation between structurally similar but functionally distinct compounds based on their higher-order interactions. In our work, our major contributions are outlined as follows:

- We develop an efficient model based on transformer architecture to identify DILI compounds.
- We combine two type of attentions: multi-head and channel-wise to enhance the predictive power.
- Our model is developed using refined dataset of DILI compounds to improve model generalizability.

## DATA COLLECTION AND FEATURIZATION

In our study, we utilized a DILI dataset comprising 1,919 molecular compounds with a slightly imbalanced class distribution (Table 1): 1,074 compounds (56%) categorized as DILI-positive (Class 1) and 845 compounds (44%) as DILI-negative (Class 0). This dataset was constructed by merging compounds from multiple sources, including *Zhang et al. (2016)*, *Ai et al. (2018)*, *Li et al. (2020)*, and *Lee & Yoo (2024)*. Duplicated samples were then systematically removed to ensure data integrity.

For feature extraction, molecular structures were processed using RDKit to generate Morgan fingerprints, circular topological fingerprints with 2,048 bits and a radius parameter of 2, effectively capturing local molecular substructure information. The featurization process converted simplified molecular input line entry system (SMILES) string representations of compounds into these fingerprint embeddings for modeling. We

**Table 1 Numbers of samples in source of data.**

| Sources of data | Number of samples |
|---|---|
| *Zhang et al. (2016)* | 1,317 |
| *Ai et al. (2018)* | 1,527 |
| *Li et al. (2020)* | 1,002 |
| *Lee & Yoo (2024)* | 1,851 |
| Ours (this study) | 1,919 |

**Table 2 Numbers of samples in each category of the used dataset.**

| Data | Number of samples | | |
| | DILI-positive (Class 1) | DILI-negative (Class 0) | Total |
|---|---|---|---|
| Training | 870 | 684 | 1,554 |
| Validation | 97 | 76 | 173 |
| Test | 107 | 85 | 192 |
| All data | 1,074 | 845 | 1,919 |

randomly allocated approximately 90% of samples for model training (with a ratio of 90:10 for training and validation data), and reserved 10% for final performance evaluation (Table 2). Finally, all data were converted to PyTorch tensors to facilitate efficient computation and compatibility with the frameworks.

# MODEL ARCHITECTURE

## Multi-head self-attention

Multi-head self-attention is one of the most fundamental attention mechanism proposed by *Vaswani et al. (2017)*. The multi-head option allows parallel processing across different representation subspaces. This mechanism extends traditional attention by computing $h$ separate attention operations simultaneously. For input $X \in \mathbb{R}^{n \times d}$, each attention head $i$ projects the input into distinct *Query*, *Key*, and *Value* spaces using learned parameters $W_i^Q$, $W_i^K$, and $W_i^V$, respectively. For each head, the learned information is expressed as:

$$\text{Head}_i = \text{Softmax}\left(\frac{(X \odot W_i^Q)(X \odot W_i^K)^T}{\sqrt{d_k}}\right)(X \odot W_i^V). \tag{1}$$

These individual head outputs are then concatenated and projected *via* $W^O$ to form the final multi-head attention output, presented as:

$$\text{Multi-head}(X) = \text{CONCAT}(\text{Head}_1, \text{Head}_2 \ldots, \text{Head}_i) \odot W^O. \tag{2}$$

The multiple heads help users to flexibly adjust the model to attend to information from different representational perspectives. While some heads can focus on local patterns, others capture long-range dependencies or semantic relationships. The adjustable number of head makes this model effectively adapt to various computing resources required. This mechanism is demonstrated to significantly enhances the model learning capacity.

## Channel-wise attention

Channel-wise attention mechanism was first introduced in Squeeze-and-Excitation (SE) networks (*Hu, Shen & Sun, 2018*). This attention mechanism provides an efficient adaptive feature recalibration during the learning process. Given an intermediate feature map $F \in \mathbb{R}^{C \times H \times W}$ with $C$ channels of height $H$ and width $W$, the channel-wise attention first applies a squeeze operation that aggregates the spatial information into a channel descriptor. This is typically implemented as global average pooling, producing a vector $z \in \mathbb{R}^C$ where each element $z_c$ is calculated as:

$$z_c = \frac{1}{H \times W} \sum_{i=1}^{H} \sum_{j=1}^{W} F_c(i,j). \tag{3}$$

The excitation operation then captures channel-wise dependencies through a simple gating mechanism with learnable parameters. This is formulated as:

$$s = \sigma(W_2 \odot \delta(W_1 \odot z)), \tag{4}$$

where $\delta$ represents a non-linear activation function (commonly ReLU), $\sigma$ denotes the Sigmoid activation, $W_1 \in \mathbb{R}^{\frac{C}{r} \times C}$ and $W_2 \in \mathbb{R}^{C \times \frac{C}{r}}$ are parameter matrices, and $r$ is a reduction ratio controlling the capacity of the bottleneck. The final output is produced by rescaling the original feature map with the activated attention weights:

$$\tilde{F}_c = s_c \odot F_c. \tag{5}$$

This mechanism allows the network to selectively emphasize informative features while suppressing less useful ones, effectively performing dynamic channel-wise feature recalibration that enhances representational power with minimal computational cost required. The output of the SE block is used as a residue added to the output of the main branch.

## Proposed model

Our proposed architecture (Fig. 1) begins with a feature expansion module that consists of two fully connected (FC) layers. The first layer increases the dimensionality of the 128-dimensional input features to 256 dimensions, followed by Batch Normalization (BatchNorm), a LeakyReLU activation function (with a slope of 0.01), and a dropout layer (with a rate of 0.2). The same pattern is repeated in the second layer, preserving the 256-dimensional representation. This design effectively creates a robust pathway for deep feature extraction and incorporates strong regularization techniques to reduce overfitting. Then, a Transformer layer is introduced to process 32-dimensional embeddings by utilizing self-attention through a multi-head attention mechanism, enabling the model to selectively focus on relevant molecular substructures. The feed-forward network within the Transformer expands the representation from 32 dimensions to 2,048 dimensions before condensing it back to 32 dimensions. Layer normalization (LayerNorm) and dropout (with a rate of 0.1) are applied throughout the Transformer block to stabilize training and improve generalization. Following the Transformer block, the model employs a third linear layer that expands the output from 32 dimensions to 128 dimensions, followed by

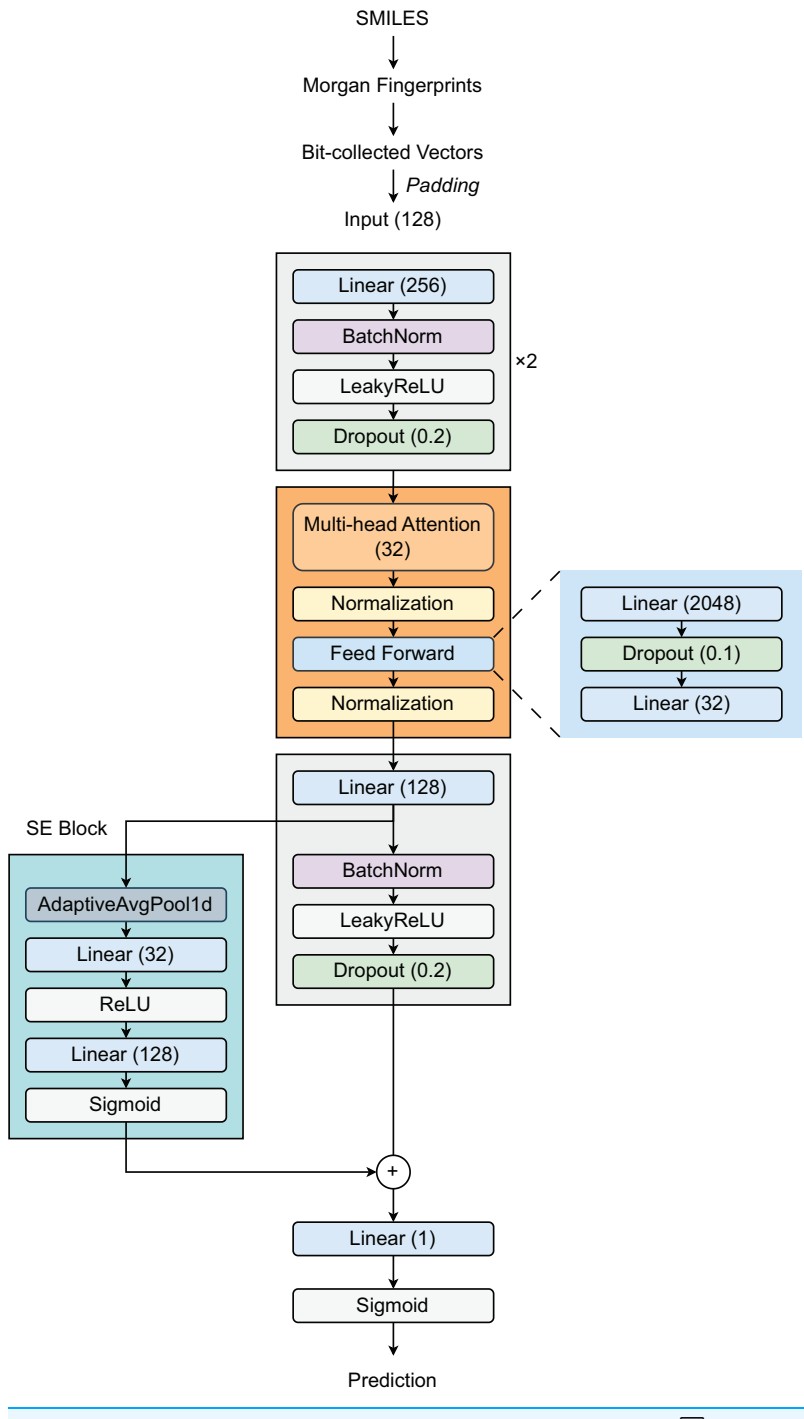

**Figure 1  Model architecture.**

BN, LeakyReLU activation, and dropout layers. The integration of a Squeeze-and-Excitation Block (SEBlock) that implements channel-wise attention helps adaptively recalibrate feature responses by explicitly modeling interdependencies between channels through a bottleneck structure with Sigmoid activation. This block allows the network to emphasize informative features. Our architecture also incorporates a residual connection

through a dedicated linear layer using a skip connection to maintain gradient flow during training. The final output layer reduces the 128-dimensional enhanced representation to a single value, appropriate for binary classification tasks such as DILI prediction.

### Model training

The model was trained using binary cross-entropy (BCE) loss for binary classification with a stochastic gradient descent (SGD) optimizer with a learning rate of 0.001, a momentum of 0.9, and a weight decay of $1e^{-5}$. We installed a learning rate scheduler to automatically reduce the learning rate when a monitored metric on the validation loss plateaus. The training process ran for a maximum of 40 epochs with early stopping (patience = 5) to prevent overfitting, using shuffled batches of 16 samples. Model performance was ultimately evaluated on a test set to assess the DILI prediction task.

### Computing resources

All modeling processes were completed on the PyTorch 2.0 framework using an NVIDIA RTX 3090 GPU with 24 GB of VRAM. All data transformation and processing stages were performed on a Windows 11 system characterized by an Intel Core i7-12700K Processor (3.60 GHz base, 5.0 GHz turbo) and 32 GB of RAM.

### Evaluation criteria

To evaluate the model performance, we used area under the curve-receiver operating characteristic (AUC-ROC), area under the curve-precision-recall (AUC-PR), and F1 Score.

## EXPERIMENTAL RESULTS

Table 3 presents a comparative analysis of our proposed deep learning architecture against machine learning baselines, including RF, SVM, and logistic regression (LR).

Table 4 demonstrates the comparative effectiveness of our proposed architecture against state-of-the-art graph neural network approaches specifically designed for molecular representation learning.

To evaluate the robustness of our proposed model, we conducted five independent trials with different random sampling seeds. Table 5 summarizes the performance metrics across these experiments.

## DISCUSSION

### Benchmarking with machine learning models

The benchmarking results in Table 3 demonstrate that iDILI-MT substantially outperforms conventional machine-learning classifiers across all evaluated metrics. Specifically, iDILI-MT attains an AUC-ROC of 0.8686, representing a significant improvement over the best-performing baseline, SVM (AUC-ROC = 0.7500). Similarly, the AUC-PR increases from 0.7931 (SVM) to 0.8869 for iDILI-MT, indicating enhanced capability in prioritizing true hepatotoxic compounds amid a predominance of non-toxic examples. The F1 score of 0.8487 further confirms that iDILI-MT maintains a favorable

**Table 3 Performance of our model benchmarked against other machine learning models.**

| Method | AUC-ROC | AUC-PR | F1 score |
| --- | --- | --- | --- |
| RF | 0.7338 | 0.7894 | 0.7463 |
| SVM | 0.7500 | 0.7931 | 0.8571 |
| LR | 0.7346 | 0.7873 | 0.7518 |
| Ours | 0.8686 | 0.8869 | 0.8487 |

**Table 4 Performance of our model benchmarked against other deep learning models.**

| Method | AUC-ROC | AUC-PR | F1 score |
| --- | --- | --- | --- |
| molecularGNN | 0.7342 | 0.8211 | 0.7705 |
| MD-GNN | 0.7202 | 0.7657 | 0.7721 |
| Molgraph | 0.8472 | 0.8886 | 0.7727 |
| DMFPGA | 0.8500 | 0.8617 | 0.7346 |
| ResNet18DNN | 0.5641 | 0.5792 | 0.7028 |
| Ours | 0.8686 | 0.8869 | 0.8487 |

**Table 5 Performance variation of our proposed model across five independent trials.**

| Trial | AUC-ROC | AUC-PR | F1 score |
| --- | --- | --- | --- |
| 1 | 0.8567 | 0.9108 | 0.8033 |
| 2 | 0.8357 | 0.8835 | 0.7965 |
| 3 | 0.8696 | 0.9021 | 0.8448 |
| 4 | 0.8310 | 0.8690 | 0.7965 |
| 5 | 0.8563 | 0.8870 | 0.8452 |
| Mean | 0.8499 | 0.8905 | 0.8173 |
| SD | 0.0161 | 0.0164 | 0.0255 |

trade-off between precision and recall, surpassing random forest and logistic regression models and matching the highest baseline performance.

These improvements can be ascribed to the hybrid architecture of iDILI-MT, which integrates a multi-head Transformer encoder with a SE attention module. The Transformer encoder enables the model to capture long-range dependencies within sequential molecular representations, thereby discerning intricate patterns spanning distal substructures. Concurrently, the SE module adaptively recalibrates channel-wise feature importance, allowing the network to emphasize chemical attributes most predictive of hepatotoxicity. Collectively, these mechanisms facilitate both global contextual understanding and fine-grained feature discrimination. Accordingly, iDILI-MT not only achieves superior predictive accuracy but also affords interpretable insights into the substructural determinants of drug-induced liver injury, rendering it a valuable asset for early safety assessment in drug discovery.

## Benchmarking with deep learning models

Table 4 compares the performance of iDILI-MT against several leading deep learning architectures: molecularGNN (*Wang, Li & Farimani, 2022*), MD-GNN (*Chen et al., 2023*), Molgraph (*Kensert, Desmet & Cabooter, 2024*), DMFPGA (*Wang et al., 2024*), and ResNet18DNN (*Chen et al., 2021*). iDILI-MT achieves an AUC-ROC of 0.8686, exceeding both Molgraph (0.8472) and DMFPGA (0.8500). In the AUC-PR metric, iDILI-MT attains 0.8869, again outperforming DMFPGA (0.8617). The most substantial improvement is observed in the F1 Score, where iDILI-MT reaches 0.8487.

Graph-based models exhibit larger performance gaps: molecularGNN records an AUC-ROC of 0.7342, AUC-PR of 0.8211, and F1 Score of 0.7705, while MD-GNN achieves 0.7202, 0.7657, and 0.7721 on the same metrics, respectively. ResNet18DNN shows the lowest results, with an AUC-ROC of 0.5641, AUC-PR of 0.5792, and F1 Score of 0.7028.

These results demonstrate that the hybrid architecture of iDILI-MT, which integrates multi-head self-attention with a squeeze-and-excitation module, more effectively captures the complex structural and chemical patterns relevant to drug-induced liver injury than traditional graph neural network approaches alone.

## Performance stability across independent trials

Table 5 summarizes the performance of iDILI-MT over five independent trials with different random seeds. The model attains a mean AUC-ROC of 0.8499 (SD = 0.0161), a mean AUC-PR of 0.8905 (SD = 0.0164), and a mean F1 score of 0.8173 (SD = 0.0255). These small standard deviations indicate that the model's predictive performance is highly stable across different random initializations.

The highest AUC-ROC (0.8696) and F1 score (0.8448) were observed in Trial 3, while Trial 1 yielded the peak AUC-PR of 0.9108. Even in the lowest-performing trial (Trial 4), iDILI-MT maintained a robust AUC-ROC of 0.8310 and AUC-PR of 0.8690. Overall, the narrow range of variation across all metrics confirms that iDILI-MT delivers consistent and reliable DILI predictions irrespective of random sampling.

## Limitations and future work

Despite the strong performance of iDILI-MT, there are several key limitations to address. First, the model currently relies solely on chemical structure inputs and does not integrate orthogonal data sources, such as biological pathway annotations or drug-metabolizing enzyme profiles, that are known to influence DILI risk. Second, although we observed stable results on our held-out test set, external validation across diverse and novel chemical scaffolds is necessary to confirm the model's generalizability.

To overcome these gaps, future work could: (1) embed multi-head attention–based recommendation mechanisms to propose safer compound alternatives in the context of drug–target interactions, drawing inspiration from recent repurposing networks (*Feng et al., 2024*); (2) leverage pretrained sequence embeddings from nucleotide language models to capture richer substructural motifs (*Li et al., 2024*); (3) extend iDILI-MT into a multi-view framework that fuses 1D SMILES, 2D graph representations, and predicted 3D

conformers for more robust molecular encodings (*Zhang et al., 2024*); (4) incorporate evolutionary-ensemble strategies to dynamically optimize attention head configurations and network architectures (*Nguyen et al., 2022*); and (5) adopt a unified multimodal contrastive learning scheme that aligns transformer-derived embeddings with geometry-aware features to heighten sensitivity to stereochemical patterns linked to hepatotoxicity (*Nguyen et al., 2024*). Together, these enhancements will move iDILI-MT toward a more comprehensive, biologically informed, and broadly applicable DILI prediction platform.

## CONCLUSION

We developed iDILI-MT, a novel computational framework for DILI prediction that leveraged a hybrid architecture incorporating multi-head transformer and squeeze-and-excitation mechanisms. Our model consistently outperformed traditional machine learning methods and state-of-the-art graph neural networks. Through multiple independent trials, we demonstrated remarkable model stability with minimal performance variation across different random seeds. The multi-level attention mechanisms effectively integrated both local and global molecular features, providing a more comprehensive understanding of structure-toxicity relationships. This work established a robust computational approach for early identification of hepatotoxic compounds, addressing a critical challenge in pharmaceutical development and drug safety evaluation.

### Funding

The study was supported by the Education and Scientific Research Project for Middle-Aged and Young Teachers in Fujian Province (JAT220750) and Fujian Province Natural Science Foundation (grant number: 2023J011900). The funders had no role in study design, data collection and analysis, decision to publish, or preparation of the manuscript.

### Grant Disclosures

The following grant information was disclosed by the authors:
Education and Scientific Research Project for Middle-Aged and Young Teachers in Fujian Province: JAT220750.
Fujian Province Natural Science Foundation: 2023J011900.

### Competing Interests

The authors declare that they have no competing interests.

### Author Contributions

- Wanrong Zheng conceived and designed the experiments, performed the experiments, analyzed the data, performed the computation work, prepared figures and/or tables, authored or reviewed drafts of the article, and approved the final draft.

- Fobao Lai conceived and designed the experiments, performed the experiments, analyzed the data, performed the computation work, prepared figures and/or tables, authored or reviewed drafts of the article, and approved the final draft.

## Data Availability

The data and code used in the experiments are available in the Supplemental Files.

## Supplemental Information

Supplemental information for this article can be found online at http://dx.doi.org/10.7717/peerj-cs.2973#supplemental-information.

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
