# Peer review of "iDILI-MT: identifying drug-induced liver injury compounds with a multi-head Transformer"

_PeerJ Computer Science, doi:10.7717/peerj-cs.2973_

## Round 0.1 · original submission · Major Revisions

Thank you for submitting your manuscript to our journal. While the reviewers recognize the potential significance of your research, they have identified substantial concerns that require extensive revisions before further consideration for publication. The reviewers have also highlighted significant gaps in their comments. These concerns need to be addressed. Given the extent of these revisions, we invite you to submit a thoroughly revised manuscript, accompanied by a detailed response addressing each point raised by the reviewers. If you choose to resubmit, your manuscript will undergo another round of review to assess how effectively the concerns have been addressed.

Reviewer 1 ·

Basic reporting

The manuscript introduces iDILI-MT, a novel hybrid architecture integrating multi-head transformer mechanisms with squeeze-and-excitation attention modules for the prediction of drug-induced liver injury (DILI). The study is well-positioned within a critical biomedical context, offering a sophisticated model that captures both local substructures and global molecular interactions. The use of Morgan fingerprints for molecular encoding and the combination of transformer attention with channel-wise recalibration represent important architectural innovations. The reported results demonstrate competitive performance over traditional ML models and GNN-based approaches, showing strong metrics in terms of AUC-ROC, AUC-PR, and F1 score.
However, there are still several issues to be tackled. The model is heavily dependent on chemical structure data and omits biological pathway features, which may limit its real-world application and biological relevance. The generalizability of the model is insufficiently explored beyond internal cross-validation, and external datasets with diverse chemical scaffolds were not used for validation. Furthermore, some technical explanations, especially within the architecture description and data processing, lack clarity and coherence. The literature review is extensive but would benefit from better synthesis and organization. Overall, while the manuscript presents valuable contributions, it requires refinement in methodology explanation and validation rigor.
1) First of all, within the introduction section, authors are advised to narrow the scope of general DILI background and emphasize the motivational gap—why existing models are insufficient and how iDILI-MT addresses this specifically. It is better to rewrite the last part of the introduction to clearly list three key technical contributions would help establish a sharper research objective.
2) For the literature review of this paper, authors should add discussion of prior works should include critical insight, such as what limitations still persist despite reported high performance (e.g., small datasets, lack of external validation).
3) For the methodology of this paper, authors are encouraged to provide a visual flowchart or pseudo-code describing the complete model pipeline, from data preprocessing to prediction, to improve accessibility. The feature expansion module and transformer block explanations can be improved by clearly separating input/output dimensions and justifying design choices, such as why a 32-dimensional Transformer is followed by an expansion to 2048.
4) Additionally, the channel-wise attention section should explicitly mention how SEBlock is integrated within the Transformer pipeline (before or after residual connection).
5) Finally, authors should also clarify whether feature order or position in the Morgan fingerprints is preserved or shuffled during positional encoding, and how this affects downstream performance. Thanks.

Experimental design

No comment.

Validity of the findings

Please refer to my basic reporting section.

Additional comments

No comment.

Reviewer 2 ·

Basic reporting

- Overall, the paper is straightforward and easy to understand.
- The authors did a commendable job of surveying related works.
- They provided supplementary materials, including data and source code; however, the evaluation code is absent. In addition, the authors should also provide the source code of the methods that they used to compare in the experiment section (or explain how to get these numbers, because they used their newly curated data).

Experimental design

The proposed method is based on the transformer architecture and utilizes Multi-head Self-Attention and Channel-wise Attention, which can help capture diverse biochemical interaction patterns, addressing both local substructural features and global molecular topologies simultaneously.

Below are some suggestions to improve the paper presentation and detail:

- The authors should provide a table comparing their collected data with the previous datasets and explaining the need for the new curated data, how it is better than the previous one, and whether it solves any drawbacks of the previous datasets.
- The authors should provide a figure of the model architecture (instead of the current Figure 1 and Proposed Model section), which shows the input, processing steps if needed, the transformer architecture, which includes layers of Multi-head Self-Attention and Channel-wise Attention, the output prediction, etc.

Validity of the findings

The authors provided experiments to validate the proposed method's performance. However, the authors should address some questions below:

- What are the RF and LR models? I can guess the RF will be Random Forest, but not the LR.
- The caption of Table 3 should be "other deep learning models" instead of "other machine learning models", which is similar to Table 2.
- Please add the reference for molecularGNN, MD-GNN, and Molgraph.
- Why are the values of the proposed method in Tables 2 and 3 different from those in Table 4? The authors should use the values in Table 4 for other tables.
- The authors should also compare with recent methods, such as DeepDILI and InterDILI, which were mentioned in the Related Work section.

Additional comments

No comment.

---

## Round 0.2 · accepted · Accept

Based on reviewers' decisions, we are delighted to inform you that your manuscript, "iDILI-MT: Identifying drug-induced liver injury compounds with a Multi-Head Transformer", has been accepted for publication in PeerJ Computer Science.

Reviewer 1 ·

Basic reporting

After carefully checked on revisions as well as feedback of authors for my revision recommendations in previous version of this paper, I confirmed all concerns have been sufficiently resolved in the latest version, as a result, I thought this paper can be accepted for publication in this form. Thanks.

Experimental design

No comment.

Validity of the findings

Please refer to my basic reporting section.

Additional comments

No comment.

Reviewer 2 ·

Basic reporting

The revised manuscript successfully aligns with the journal's established standards and guidelines. After a thorough review, I have no additional comments or suggestions.

Experimental design

The authors have thoroughly revised the manuscript, taking into account our feedback. They have also conducted additional experiments to address the specific points we raised. After reviewing the changes and the new data presented, I have no further comments or concerns to add.

Validity of the findings

The experiments and evaluations were conducted with satisfactory results, demonstrating their reliability and effectiveness. The discussion section has been significantly improved, with a more thorough analysis of the methodology and an in-depth examination of its limitations. This includes a critical assessment of potential biases, constraints in the experimental design, and suggestions for future studies. At this point, I have no additional comments to contribute.

Additional comments

No comment.